# Impact of Gender on STEAM Education in Elementary School: From Individuals to Group Compositions

**DOI:** 10.3390/bs12090308

**Published:** 2022-08-26

**Authors:** Lin Ma, Heng Luo, Xiaofang Liao, Jie Li

**Affiliations:** Faculty of Artificial Intelligence in Education, Central China Normal University, Wuhan 430079, China

**Keywords:** gender, gender groups, STEAM, collaborative learning, quasi-experiment

## Abstract

Gender differences are essential factors influencing collaborative learning at both individual and group levels. However, few studies have systematically investigated their impact on student performance in the innovative context of STEAM education, particularly in the elementary school setting. To address this research need, this study examined the learning behaviors of 91 sixth graders in a STEAM program, who were classified into three gender groupings, namely, boy-only, girl-only, and mixed-gender groups, and further compared their performance in terms of cognition, interaction, and emotion by both gender and gender group type. The results show that, compared to individual gender differences, the gender group type had a greater impact on students’ behavioral performance during STEAM education. While all gender groupings had specific advantages, mixed-gender groups proved to be the most preferable, with benefits such as enhanced higher-order thinking, interaction, and emotional expression. Moreover, the study revealed that both boys and girls acted differently when working with the opposite gender in mixed-gender groups. These research findings have several implications for facilitating STEAM learning in co-ed elementary schools.

## 1. Introduction

In K-12 education, STEAM is defined as an interdisciplinary curriculum that integrates the five disciplines of science (A), technology (T), engineering (E), art (A), and mathematics (M) [1]. Interdisciplinary, collaborative, and realistic problem solving are the main features of STEAM education. Correspondingly, STEAM education has the benefits of developing students’ innovation abilities, cooperation abilities, and realistic problem-solving abilities [2,3,4]. Due to the benefits of STEAM education, it has been regarded as one of the most promising education methods in the world since its inception. Numerous studies have explored factors that potentially influence STEAM education. Tseng et al. (2013) explored the impact of an interactive atmosphere on students’ participation in STEAM learning [5]. Lamb et al. (2015) focused on the roles of cognitive, emotional, and content aspects in successful STEAM education [6]. Taylor and Baek (2019) proposed that group role assignments in STEAM education benefit students’ learning [7].

However, little research has been conducted on STEAM education that takes students’ gender and gender grouping into account, especially at the elementary level. Gender plays an important role in education. Gender is associated with psychological differences between boys and girls, and these differences affect their behavior and performance. Pomerantz et al. (2001) proposed that girls are psychologically more motivated than boys to please adults (e.g., teachers and parents), and this may underlie gender differences [8]. Girls interpret failure as disappointing adults, which makes them work harder to improve their grades. Boys, however, are not concerned with pleasing adults and interpret failure as being related to a particular discipline. These differences make girls more prosocial than boys, showing more positive emotions and internal emotions (e.g., sadness, anxiety, and sympathy), while boys show more external emotions (e.g., anger) [9], resulting in boys engaging in more aggressive or destructive behavior than girls [10].

Despite the proven psychological and behavioral differences in gender, there are mixed findings regarding their impact on learning. While some studies have reported that boys demonstrate superior abilities in science, reasoning abilities, and abstract knowledge [11,12], girls have been found to outperform boys in speech and reading comprehension [11,13]. Other studies have found that girls outperform boys in most subjects in K-12 education [14,15,16]. These inconsistent conclusions may be related to the fact that gender differences are influenced by age, relationships, and field [9,17]. Students in grade 6 (age 11–12) of elementary school are about to enter adolescence and have certain logical thinking and teamwork abilities. However, few studies have explored gender differences in STEAM education for this particular group of students.

Furthermore, some research studies have revealed that the gender composition of groups also leads to different learning performances during collaborative learning. General studies have concluded that mixed-gender groups perform better than same-gender groups [18,19]. This may be because during group interaction, boys talk more about task-related topics while girls are better at planning and communication [20], which makes mixed-gender groups more relaxed and prone to engaging in cooperative behavior than same-gender groups. Some studies suggest that same-gender groups are better, as they are more purposeful and consensual than mixed-gender groups [18,21]. Other people believe that the influence of group gender composition is limited and that it only affects students’ attitudes rather than their performance [22,23].

In addition, in terms of boy and girl students’ performance in different gender groupings, boys’ performance in mixed-gender groups is significantly better than that in same-gender groups, and boys prefer mixed-gender groups [22]. This may be because, in mixed-gender groups, boys are more likely to demonstrate leading behaviors, while girls are more likely to be agreeable [24,25]. According to Walker et al. (1996), in the absence of an appointed leader, boy students are 5 times more likely than girl students to assume the role of opinion leader [26], which makes them more confident and more willing to be in mixed-gender groups. Girls, however, prefer same-gender groups [22], because girls in girl-only groups have more in common with their female companions, making it easier to reach a mutual understanding and agreements. This, in turn, may improve their team effectiveness and productivity [22].

In conclusion, since STEAM education creates a stressful environment that relies more on team communication and collaboration than traditional modes of education [27], gender and the gender composition of groups tend to have a large impact on student behavior in STEAM education. However, there are insufficient research studies investigating the effects of gender and group gender composition on student learning behavior in the educational context of STEAM. Specifically, this study sought to answer the following three questions:(1)Do gender differences affect students’ learning behavior in STEAM education?(2)Do different gender groupings affect students’ learning behavior in STEAM education?(3)Do girls and boys perform differently in same- and mixed-gender groups during STEAM education?

## 2. Method

### 2.1. Research Design

This study utilized a quasi-experimental design to examine the impact of gender and gender grouping difference on students’ learning behaviors in a STEAM education program. The independent variables of this study are individual gender and gender group composition. The former is a demographic attribute that divided students naturally into two conditions: male and female, and the latter was manipulated to formulate three conditions: boys-only, girls-only, and mixed-gender (gender ratio approximately 1:1). The dependent variables are STEAM learning behavioral performance as measured by higher-order thinking, emotional expression, interaction, and irrelevant behaviors. Additionally, post-hoc comparisons were conducted to determine difference in boy and girl performance between same-gender and mixed-gender conditions.

### 2.2. Participants

A total of 91 sixth graders (age 11–12) from an elementary school in central China participated in this study. With the assistance of a school teacher, informed consent forms were distributed to children and their parents one day before the first STEAM class. Only the students who handed in the consent forms with parental approval were allowed to participate in this study. The research protocol was evaluated and approved by the Ethics Committee of the Central China Normal University (protocol code ccnu-IRB-202111047, approved on 11/11/2021).

The participants came from two classes, with 48 students in the first class and 43 students in the second class. Among them, five participants did not participate in the whole experiment due to time conflicts, so the final count was 86 (47 boys and 39 girls). The participants were between the ages of 11 and 12. They all came from nearby communities and had similar family economic backgrounds. Participants were randomly divided into three types of gender grouping: five boy-only groups, seven mixed-gender groups, and three girl-only groups. Most groups comprised five or six participants, and the numbers of boys and girls in the mixed-gender groups were kept as equal as possible. The distribution of boys and girls in the groups is shown in Table 1.

### 2.3. Research Context and Procedure

Before the start of the STEAM class, the research team placed a camera and a microphone in the best shooting position for each group and read out instructions to the participants, informing them that the activities in this STEAM class would be filmed but that their performance and learning outcomes would not affect their grades in the course. The group distribution in the classroom and the video recording position of the camera are shown in Figure 1.

The STEAM course comprised three lessons, with one lesson per week, and these lessons were completed within three weeks. Classroom behavior was recorded by the cameras, while four researchers observed one to two groups each. In the first lesson, participants mainly learned scientific knowledge related to the eyeball. Group activities included a discussion and a design of a myopia questionnaire. The second lesson mainly concerned mathematics and technology. Group activities included using an Excel spreadsheet to analyze data and create statistical graphs. The third lesson was poster production, which mainly dealt with artistic skills. The groups needed to work together to present the research findings visually in a poster. Participants worked in groups throughout the STEAM learning process.

### 2.4. Data Collection

The primary data collected in this study were video recordings of the participants’ classroom behaviors. We coded the video recordings using the theoretical framework of learning engagement proposed by Fredricks et al. (2004), quantifying participants’ performance into three categories: cognition, behavior, and emotion [28]. The detailed coding protocol is listed in Table 2.

The cognitive dimension records instances of higher-order thinking as defined by Bloom’s taxonomy of cognitive learning objectives [29], which includes three observable behaviors: analysis, application, and evaluation. These three behaviors were selected due to their prevalence in STEAM education and relevance to cognitive engagement.

The codes for the emotional dimension were informed by the sociology of emotion proposed by Stets (2010) who classified emotions into the dichotomic categories of positive and negative emotions [30]. Table 2 lists the common types of emotion expressions observed during STEAM learning and assign them into the proper categories.

The behavioral dimension focuses on social interaction and the relevance of learner action. The former emphasizes the social constructivist nature of STEAM [31] whilst the latter indicates the behavioral engagement of the participants [28]. Interaction behaviors comprise verbal interaction, nonverbal interaction, listening, leading behavior, and hand raising; and irrelevant behaviors comprise irrelevant discourse, irrelevant action, and destructive behavior. Apart from leading behaviors, most of the above-mentioned behaviors are self-explanatory. The coding of leading behaviors was informed by the works of Li et al. (2007) [32].

The video capture device used was DJI POCKET 2, which has 64 megapixels and a 4K resolution, and it works with a microphone to ensure that both video and sound are sufficiently clear. The video coding process includes two phases. In the first phase (January–February, 2022), the first, third, and fourth authors divided the video recordings into 2-min segments as units of analysis because most learning events can be captured in such segment length and the amount of workload is manageable for manual coding. The total length of video is 225 min, which was divided into 113 coding segments. Using the coding framework described in Table 2, the three authors freely coded approximately 30% of the video segments with the purpose of further validating and revising the tentative coding protocol. After finalizing the coding protocol, we initiated the second phase (March–April, 2022), where 12 undergraduate students were recruited as volunteers to code the video segments. The volunteer coders all took rigorous coding training for 4 h and passed the coding assessment based on a 30-min sample video. Including the three authors, a total of 15 persons jointly participated in the coding process, and each video segment was coded by two persons to ensure inter-rater reliability. Any controversial issues and disagreement emerged in the coding process were resolved through weekly discussions among the research team. Upon reaching satisfactory reliability, the mean score of the two coders would be used as the final coding statistics.

### 2.5. Data Analysis

Spearman’s correlation coefficient test was performed to measure inter-rater reliability, and major discrepancies in coding were discussed and recoded. The final correlation coefficient of each data point was maintained between 0.72 and 0.94. According to De Winter et al. (2016), a Spearman’s correlation coefficient greater than 0.7 is considered to be a strong correlation [33], so the coded data have good reliability and the quantitative results are reliable.

After obtaining coded data with good reliability, a nonparametric test was conducted on the coded data because the coded data did not conform to normality and homogeneity. Gender and gender grouping were used as independent variables, and coding behavior was used as the dependent variable. Additionally, we further compared participant behaviors in same-gender groups and mixed-gender groups to examine how group composition affected individual learning performance. Mann-Whitney U test was used to determine difference between two experiment conditions (e.g., boys and girls, same-gender and mixed-gender groups), and Kruskal-Wallis test was conducted to determine difference among three groups (e.g., all-boys, mixed-gender, and all-girls groups) with Dunn’s test further performed as post-hoc pairwise comparison. The data analysis software used was IBM SPSS software (version 20).

## 3. Results

Gender and gender grouping were the two independent variables in this study, and the dependent variables were participants’ interaction behaviors, irrelevant behaviors, higher-order thinking behaviors, and emotions. The means and standard deviations of all independent variables for the boys, the girls, and the three types of gender grouping are listed in Appendix A. To further explore how the presence of the opposite gender affected individual learning, the means and standard deviations of participant behavior in both same-gender and mixed-gender groups are listed in Appendix B.

### 3.1. Boys vs. Girls

The main results are shown in Figure 2. In general, there were no significant differences in the interaction behaviors between boys and girls, except for the behavior of hand raising (*MD* = 6.37***, *U* = 432.000, *Z* = −4.648, *p* = 0.000). However, boys demonstrated significantly more irrelevant behaviors than girls, including irrelevant discourse (*MD* = 2.61**, *U* = 663.500, *Z* = −2.743, *p* = 0.006), irrelevant action (*MD* = 4.38**, *U* = 649.000, *Z* = −2.856, *p* = 0.004), and destructive behavior (*MD* = 0.58***, *U* = 669.500, *Z* = −3.767, *p* = 0.000). Appendix A lists the means and standard deviation values for all behaviors.

### 3.2. Girl-Only Groups vs. Boy-Only Groups vs. Mixed-Gender Groups

The influence of gender grouping on STEAM learning seemed to be greater than that of gender. The main results are shown in Figure 3. The mixed-gender groups generally exhibited more interaction behaviors than the same-gender groups. What stands out in the figure is that the mixed-gender groups engaged in significantly more verbal interactions than both the boy-only groups (*MD* = 17.87**, *p* = 0.003) and the girl-only groups (*MD* = 20.25**, *p* = 0.006). The girl-only groups engaged in significantly more nonverbal interactions than the boy-only groups (*MD* = 8.18**, *p* = 0.008), and the mixed-gender groups were in between. In terms of listening, the boy-only groups performed significantly better than the girl-only groups (*MD* = 5.37*, *p* = 0.018). There were no significant differences between the three gender grouping types in leadership behavior, but the mixed-gender groups performed the best. In terms of hand raising, the girl-only groups exhibited this behavior the least, less frequently than the boy-only groups (*MD* = −6.08*, *p* = 0.012) and the mixed-gender groups (*MD* = −6.09*, *p* = 0.058).

In terms of irrelevant behaviors, there were no significant differences between the three types of gender grouping in irrelevant actions and irrelevant discourse, but the mixed-gender groups exhibited the most irrelevant actions, and the girl-only groups exhibited the fewest irrelevant actions. Furthermore, the mixed-gender groups engaged in irrelevant discourse the least, and the boy-only groups engaged in irrelevant discourse the most. The boy-only groups exhibited significantly more destructive behaviors than the girl-only groups (*MD* = 0.71*, *p* = 0.012).

In terms of higher-order thinking behaviors, the mixed-gender groups performed better than the same-gender groups. The mixed-gender groups engaged in more analysis behaviors, which was significantly different from the girl-only groups (*MD* = 4.23**, *p* = 0.002). The mixed-gender groups exhibited more application behaviors than the girl-only groups (*MD* = 3.94***, *p* = 0.000) and the boy-only groups (*MD* = 3.22**, *p* = 0.002), and the mixed-gender groups exhibited more evaluation behaviors than the girl-only groups (*MD* = 3.85***, *p* = 0.000).

Finally, in terms of emotions, the mixed-gender groups exhibited more positive emotions overall, but the only significant differences were found in comparison with the girl-only groups (*MD* = 12.15**, *p* = 0.001). Interestingly, the mixed-gender groups also had the most negative emotions, which was significantly different from both the boy-only groups (*MD* = 12.06*, *p* = 0.013) and the girl-only groups (*MD* = 16.17**, *p* = 0.001).

### 3.3. The Influence of Group Gender Composition on Gender Performance

#### 3.3.1. Boys in Boy-Only Groups vs. Boys in Mixed-Gender Groups

There were also many differences in how boys behaved in different types of gender grouping. The main results are shown in Figure 4. In terms of interaction behaviors, the boys in the mixed-gender groups engaged in more verbal interactions (*MD* = 17.53*, *U* = 178.500, *Z* = −2.477, *p* = 0.013) than those in the boy-only groups, but they listened less (*MD* = −6.22**, *U* = 161.000, *Z* = −2.823, *p* = 0.005) than those in the boy-only groups. There was no significant difference in nonverbal interaction, leading behavior, or hand raising. In terms of irrelevant behaviors, the boys in the mixed-gender groups demonstrated more irrelevant actions than those in the boy-only groups *(MD* = 8.99**, *U* = 155.500, *Z* = −2.933, *p* = 0.003), but there was no significant difference in irrelevant discourse or destructive behavior. In terms of higher-order thinking behaviors, the boys in the mixed-gender groups displayed more application *(MD* = *3.6**, U* = 149.500, *Z* = −3.061, *p* = 0.002) and evaluation *(MD* = 3.33**, *U* = 168.500, *Z* = −2.681, *p =* 0.007) behaviors than those in the boy-only groups, but there was no significant difference in analysis behaviors. In terms of emotions, the boys in the mixed-gender groups exhibited more positive emotions, yet this difference was statistically insignificant. The boys in the mixed-gender groups exhibited more negative emotions than those in the boy-only groups (*MD* = 8.03***, *U* = 123.000, *Z* = −3.579, *p* = 0.000).

#### 3.3.2. Girls in Girl-Only Groups vs. Girls in Mixed-Gender Groups

Similarly to how boys behaved differently across group types, girls also performed significantly differently across gender groupings. The main results are shown in Figure 5.

In terms of interaction behaviors, the girls in the mixed-gender groups engaged in more verbal interactions (*MD* = 20.59**, *U* = 91.500, *Z* = −2.926, *p* = 0.003) and listened more (*MD* = 5.3**, *U* = 103.500, *Z* = −2.603, *p* = 0.009) than those in the girl-only groups, but there were no significant differences in nonverbal interaction, leading behavior, or hand raising. In terms of irrelevant behaviors, there were no significant differences overall; the girls in the mixed-gender groups showed more irrelevant actions compared to the girls in the girl-only groups (*MD* = 2.29, *U* = 186.500, *Z* = −0.355, *p* = 0.723), yet the difference was statistically insignificant. In terms of higher-order thinking behaviors, the girls in the mixed-gender groups engaged in more analysis (*MD* = 4.25**, *U* = 88.500, *Z* = −3.012, *p* = 0.003), application (*MD* = 3.55**, *U* = 73.500, *Z* = -3.423, *p* = 0.001), and evaluation (*MD* = 3.18**, U = 99.000, *Z* = −2.748, *p* = 0.006) behaviors than those in the girl-only groups. In terms of emotion, the girls in the mixed-gender groups had higher positive and negative emotions than those in the girl-only groups, but only positive emotions were significant (*MD* = 10.04**, *U* = 81.000, *Z* = −3.213, *p* = 0.001).

## 4. Discussion and Conclusions

### 4.1. Do Gender Differences Affect Students’ Learning Behavior in STEAM Education?

The first research question sought to determine whether gender contributes to differences in learning behaviors among students in STEAM education in the upper elementary grades. In contrast to previous studies that indicated a substantial influence of gender on individual learning, this study found that gender influence was only significant in a few behaviors, such as hand raising, irrelevant discourse, irrelevant actions, and destructive behaviors. A possible explanation for this might be that in the upper elementary grades, the learning tasks are simpler, and the differences between boys and girls are not yet fully revealed. Another possible explanation is that boys and girls at the ages of 11–12 years show inherently small gender differences in learning behaviors [34].

The fact that boys demonstrated more hand raising behaviors in STEAM classroom is not surprising since extroversion is often encouraged in boys rather than girls in Chinese culture, with certain characteristics such as courage and leadership valued more in male students [35]. Consequently, boys tended to be more active in the classroom, resulting in more expressive behaviors. This finding also supports the evidence from the study by Walker et al. (1996) in the U.S. context which revealed that men are five time more likely to express their opinions in mixed-gender settings than women [26]. Unfortunately, in this study, certain expressive discourses and actions were considered irrelevant to the learning task. In addition, boys exhibited more destructive behavior than girls, a finding that reflects that of Archer et al. (2011), who also found that boys exhibited more physical aggression than girls [36], and increased physical aggression is known to elicit more destructive behavior in the classroom [37].

### 4.2. Do Different Gender Groupings Affect Students’ Learning Behavior in STEAM Education?

Regarding the second research question, we found that group gender composition differences had a more significant impact on student learning in STEAM than individual gender differences. Overall, each of the three different gender groupings had its specific advantages. The mixed-gender groups stimulated more interaction behaviors and higher-order thinking than the same-gender groups, and they tended to be more expressive of their emotions (both positive and negative) during the learning process. One possible reason for this is that mixed-gender groups tend to have a more relaxed atmosphere [18], which is known to promote desirable academic, behavioral, and social–emotional outcomes for children and adolescents [38]. The boy-only groups listened to their peers more frequently during collaboration than students in the other groups, but this also led to inadequate interaction within the groups. Generally, the boy-only groups demonstrated higher goal congruence: when one group member stated a solution that everyone agreed on, the other members listened and obeyed. This finding supports the view of Jiang et al. (2017) in that members of same-gender groups are more likely to reach consensus [18]. As for the girl-only groups, they had the best discipline, with few destructive behaviors observed. This observation aligns with the common perception that Chinese girls tend to be quiet and obedient [35]. Despite the unique benefits of each gender grouping type, our findings in general favor mixed-gender grouping for improved social dialogue and cognitive engagement. Thus, we recommend mixing boys and girls in STEAM education if possible, but also urge teachers to harness the benefits of same-gender grouping when gender diversity becomes infeasible.

### 4.3. Do Girls and Boys Perform Differently in Same- and Mixed-Gender Groups during STEAM Education?

The third research question further explored whether a difference exists in the performance of students in same-gender and mixed-gender groups. The research findings revealed that both boys and girls tended to learn better in mixed-gender groups, and the presence of the opposite gender in the group benefited girls’ learning to a larger extent. The girls in mixed-gender groups became more active, demonstrating more thinking, interaction, and emotions than those in the all-girl groups. It is clear that the lively and disobedient personalities of certain boys in the groups had an influence on the girls’ perceptions of the accepted behavioral norms in STEAM education, and, thus, the increased level of activity is likely the result of social observation and modeling [39].

As for boys, mixed-gender groups also contributed to some extent to more interaction and higher-order thinking behaviors, but at the same time, mixed-gender groups also led to more irrelevant actions in boys. These non-classroom-related actions may be a way for boys to receive attention from their girl counterparts [40]. Moreover, in terms of emotions, the mixed-gender groups produced more negative emotions in the boys as opposed to the more positive emotions produced in the girls. This may be because boys tend to have higher self-esteem than girls [41], and they clearly showed frustration when they were criticized by girl members in the group for their irrelevant actions or misbehaviors.

### 4.4. Implication for STEAM Education Practice

Three implications for gender-grouping strategies in STEAM educational practice can be drawn from the present research findings. First, teachers need to create more opportunities for girls to express themselves in group work and class activities. For example, teachers could purposefully select more girls when naming students to answer questions, making the gender ratio of selectees as equal as possible. Second, teachers should pay special attention to boys’ destructive actions, such as bickering, fighting, and vandalism. A code of conduct should be reinforced to prevent such behaviors. Third, teachers should try to create mixed-gender groups rather than same-gender groups when facilitating collaborative learning in STEAM, since boys and girls in collaboration with the opposite sex are more engaged in terms of cognition, interaction, and emotion.

### 4.5. Limitations and Future Research

This study has several limitations that provide avenues for future research. First, this experiment was a quasi-experimental design in a natural learning environment, and threats to internal validity cannot be fully eliminated. For instance, although random assignment was performed, the individual difference among the participants might not be fully eliminated. Second, this study was a one-time study in the specific context of STEAM education for upper-grade students in an elementary school. Therefore, the findings may not be applicable to different age groups or collaborative learning contexts. Finally, the data analysis was based on the coding results of classroom observations but lacked in-depth narrative evidence such as interviews to support the triangulation and interpretation of the statistical results. As a result, we recommend future researchers to consider a more rigid matched-group experimental design; to replicate the study in more diverse STEAM contexts; and to focus on a variety of empirical data, including observations, interviews, and document analyses.

## Figures and Tables

**Figure 1 behavsci-12-00308-f001:**
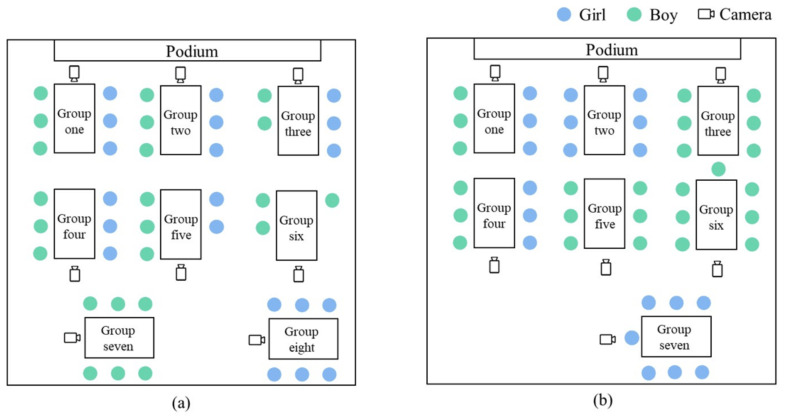
Group distribution and camera location map. (**a**) Eight groups in Class One; (**b**) Seven groups in Class Two.

**Figure 2 behavsci-12-00308-f002:**
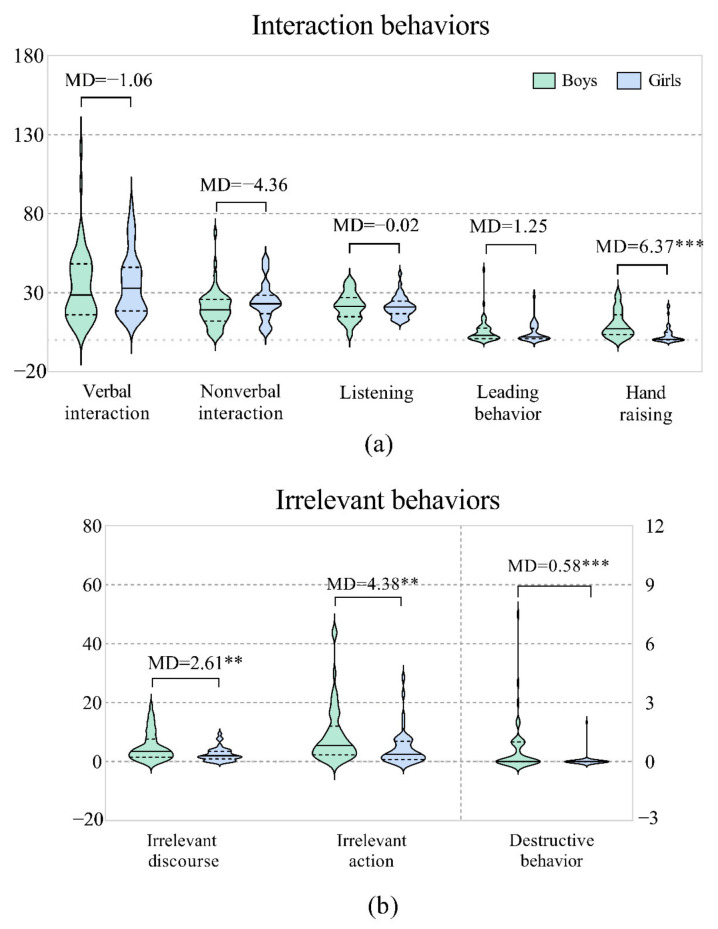
Differences in gender. (**a**) Interaction behaviors: the frequencies of verbal and nonverbal interactions are reflected on the left *Y* axis; the frequencies of behaviors of listening, leading, and hand raising are reflected on the right *Y* axis. (**b**) Irrelevant behaviors: the frequencies of irrelevant discourse and irrelevant action are reflected on the left *Y* axis, and the frequency of destructive behavior is reflected on the right *Y* axis. Note: **: *p* < 0.01; ***: *p* < 0.001.

**Figure 3 behavsci-12-00308-f003:**
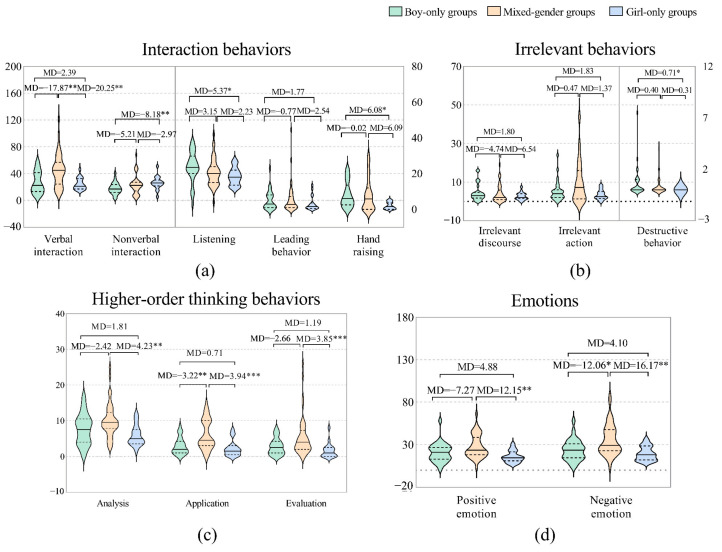
Differences in gender groups. (**a**) Interaction behaviors: the frequencies of verbal and nonverbal interactions are reflected on the left *Y* axis; the frequencies of behaviors of listening, leading, and hand raising are reflected on right *Y* axis. (**b**) Irrelevant behaviors: the frequencies of irrelevant discourse and irrelevant action are reflected on the left *Y* axis, and the frequency of destructive behavior is reflected on the right *Y* axis. (**c**) Higher-order thinking behaviors: the frequencies of analysis, application, and evaluation behaviors. (**d**) Emotions: the frequency of positive emotions and negative emotions. Note: *: *p* < 0.05; **: *p* < 0.01; ***: *p* < 0.001.

**Figure 4 behavsci-12-00308-f004:**
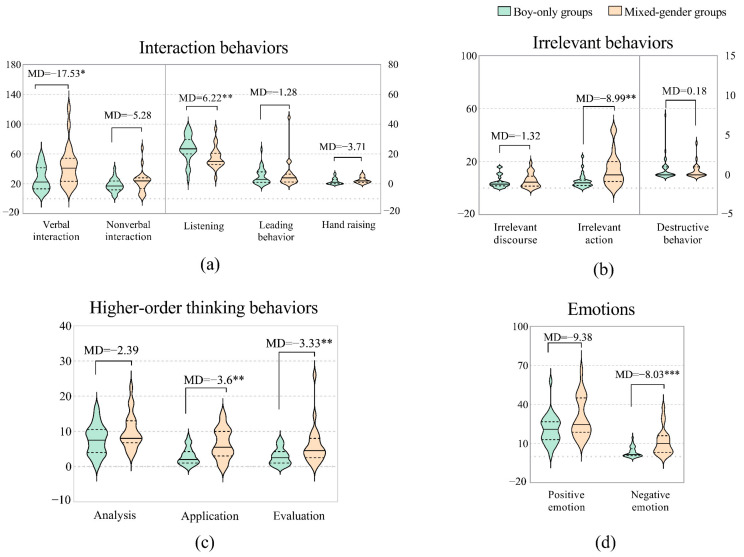
Differences between boys in boy-only groups and mixed-gender groups. (**a**) Interaction behaviors: the frequencies of verbal and nonverbal interactions are reflected on the left *Y* axis; the frequencies of behaviors of listening, leading, and hand raising are reflected on right *Y* axis. (**b**) Irrelevant behaviors: the frequencies of irrelevant discourse and irrelevant action are reflected on the left *Y* axis, and the frequency of destructive behavior is reflected on the right *Y* axis. (**c**) Higher-order thinking behaviors: the frequencies of analysis, application, and evaluation behaviors. (**d**) Emotions: The frequency of positive emotions and negative emotions. Note: *: *p* < 0.05; **: *p* < 0.01; ***: *p* < 0.001.

**Figure 5 behavsci-12-00308-f005:**
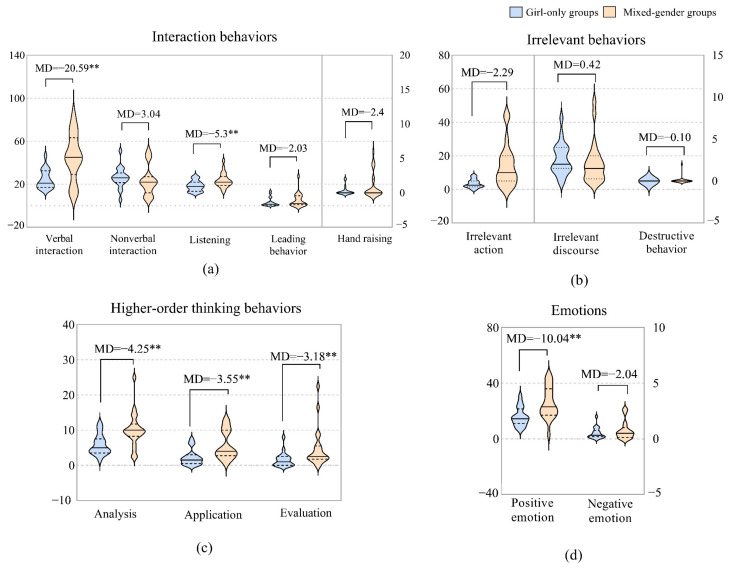
Differences between girls in girl-only groups and mixed-gender groups. (**a**) Interaction behaviors: the frequencies of verbal and nonverbal interactions are reflected on the left *Y* axis; the frequencies of behaviors of listening, leading, and hand raising are reflected on right *Y* axis. (**b**) Emotions: the frequency of positive emotion and negative emotion. (**c**) Irrelevant behaviors: the frequency of irrelevant action is reflected on the left *Y* axis, and the frequencies of irrelevant discourse and destructive behavior are reflected on the right *Y* axis. (**d**) Higher-order thinking behaviors: the frequencies of analysis, application, and evaluation behaviors. Note: **: *p* < 0.01.

**Table 1 behavsci-12-00308-t001:** The distribution of boys and girls in groups.

	Boy-Only Groups (5)	Girl-Only Groups (3)	Mixed-Gender Groups (7)	Total (15)
Boys	27	0	20	47
Girls	0	19	20	39
Total	27	19	40	86

**Table 2 behavsci-12-00308-t002:** Coding protocol used in this study.

Dimensions	Codes	Description
Cognitive	Higher-order thinking behaviors	
	Analysis	Identify the problem, justify decisions, make logical reasoning, etc.
	Application	Apply subject knowledge or technical skills to solve problems
	Evaluation	Comments and gestures of approval/disapproval/feedback
Emotional	Positive emotions	Expression of joy, pride, interest, confidence, and excitement
	Negative emotions	Expression of shame, boredom, frustration, and anger
Behavior	Interaction behaviors	
	Verbal interaction	Communication through oral conversation
	Nonverbal interaction	Communication through writing, gestures or eye contact.
	Listening	Listen attentively to peers during group work
	Leading behavior	Plan, organize, opinion seeking, task facilitation, management
	Hand raising	Actively raise hand to answer the teacher’s question or request
	Irrelevant behaviors	
	Irrelevant discourse	Chitchat unrelated to group work, off-topic discussion
	Irrelevant action	Actions unrelated to group work, distracted behaviors
	Destructive behavior	Sabotage group work, quarrel, fight, disrespectful behaviors, etc.

## Data Availability

The data presented in this study are available on request from the corresponding author. The data are not publicly available due to anonymity requirement.

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
