# Peer review of "Impact of Gender on STEAM Education in Elementary School: From Individuals to Group Compositions"

_behavsci, 2022, doi:10.3390/bs12090308_

Round 1
Reviewer 1 Report
This is an interesting study that explores the role of gender in STEAM education. The topic is relevant to the readership of the Behavioral Sciences. The manuscript was well organized and prepared. The study features a typical quasi-experimental research design. The study is clearly articulated, the results make sense, and the authors interpret them decently. The findings of the study may contribute to the design, development, and implementation of effective STEAM instruction in elementary classrooms.
There are some places that the authors could clarify to improve the manuscript for possible publication.
First, the coding scheme shown in Table 2 is a key instrument for this study and it is suggested to better clarify the specific codes. For example, what is leading behavior? What does destructive behavior mean? The authors could include a definition for each specific code (or even provide examples).
The authors mentioned that the coding scheme is based on Fredricks et al. (2004)’s engagement framework. But it was not clear how the thirteen codes were produced. Are they from any existing coding schemes, or did the authors develop them? If the latter, more information is needed on how the codes were created from the data sets.
Second, more clarity is needed for the coding process. For example, the authors mentioned that they used 2-minute segments of the collected video for analysis. Is the 2-minute video the unit of analysis? How many 2-minute segments did they analyze? Did they analyze all the videos they collected or only some sampled segments? Also, more details are needed for the specific coding steps they took. How did they use the coding scheme to approach the data? Did they code the data adaptively- were there any new incidents that emerge from the data that could add new and important properties to the coding scheme? Did they provide any training to the 15 coders?
Third, for the Discussion and Conclusions, the authors summarized the results they obtained and explained the potential reasons for the findings. The discussion will be stronger if the authors could discuss more what do the specific findings mean for future research and educational practices. For example, the authors found that the “group gender composition differences had a more significant impact on student learning”, then what important insight we could get from this finding for designing and developing effective STEAM instruction? Also, instead of describing a local problem, how the results could be linked to the interest of the whole community.
Additionally, the authors may consider including more recent studies on the gender difference in STEAM learning. The characteristics of a certain gender group (e.g. boys exhibit more aggression than girls, “men are more willing to express themselves than women”) might be significantly different from what they showed ten or twenty years ago.
Reviewer 2 Report
The manuscript is interesting considering the current times. However, there are several areas that need revision prior to publication as listed below.
i. Quasi-experiment is stated in the list of Keywords, and Limitations and Future Research, but there is no further elaboration of quasi-experimental design in the writing of research methodology. The methodology would read better if there is further elaboration of the research design used
ii. There was no description on what type of non parametric tests were used to determine a significant difference between two independent groups (gender) and three independent groups (gender grouping).
iii. It is suggested to use a standard way of reporting non parametric test results regarding the significant difference between the groups (e.g. APA style)
iv. The authors should explain how the informed consent was obtained from participants before the participants’ classroom behaviors were recorded.
Reviewer 3 Report
First of all, thank you for inviting me to review the paper entitled “Impact of Gender on STEAM Education in Elementary School: From Individuals to Group Compositions”
The study is well conceived and paper is well written.
Some minor recommendations and revisions are provided.
1. Although the students are randomly assigned to groups, how did you account for the individual differences of the students?
2. Since the students are minor, how about parents ‘approval?
3. With regards to the 15 individuals who watched the videos, this should be elaborated. Were training provided? what are the selection criteria? Are they compensated? The coding process should be further elaborated, since this is the main data collection process.
4. Fredricks et al. (2004) was used as framework for the coding of videos, while looking into cognitive, behavioral, and emotional component of the group interaction. This should be further strengthen and explained. Although some definitions are provided, however, as this is the core of your study, literature review should be accomplished.
5. Were you able to consider a pre-post test design?
In sum this study is quite interesting only some minor clarifications and recommendations needed.
